# ADDRESSING REAL-TIME FRAGMENTARY INTERACTION CONTROL PROBLEMS VIA MULTI-STEP REPRESENTATION REINFORCEMENT LEARNING

## ABSTRACT

Fragmentary interaction control problem is common in real-time control scenarios. For example, the delay or the loss of the network packets (caused by network obstacles, inadequate bandwidth, or switch faults) will lead to dynamic interval or fragmentary interaction. Moreover, fragmentary interaction hinders the application of reinforcement learning algorithms in real-time control tasks: when the states are not received, the reinforcement learning (RL) algorithm cannot make the decision for the agent according to the traditional MDP, which leads to the standstill of the agent, and finally leads to low efficiency or even failure in completing the task. However, such problems are not well studied in the RL community. In this paper, we propose to simultaneously generate multiple actions for future states in case some future states cannot be perceived. We present **M**ulti-step **A**ction **R**epre**S**entation (**MARS**) to learn a compact and decodable latent space for the original multi-step action space. Besides, our method enhances the environmental dynamic semantics of the action representation through unsupervised environmental dynamics prediction and action transition scale. Based on MARS, the RL algorithms optimize policies in the learned representation space and interact with the environment by decoding the latent actions to the original ones. MARS outperforms the existing state-of-the-art baselines in a variety of fragmentary interaction real-time control tasks. Further, MARS significantly improves the performance of high-frequency robot control tasks based on fragmentary interaction in the real-world.

## 1 INTRODUCTION

In recent years, the field of **d**eep **r**einforcement **l**earning (DRL) has witnessed striking empirical achievements in a variety of Markov Decision Process (MDP) problems (Mnih et al., 2013; Kaufmann et al., 2023). However, when applying RL algorithms to real-time (especially remote) control tasks, we will face an inevitable challenge: the delayed or fragmentary interaction issue induced by communication latency or packet loss. In a real-time control task, the interaction between the action executor and the agent is conveyed over a channel using data packets. When the packet is lost due to link failure[1], the RL agent cannot make decisions, and the action executor will abruptly stop or simply repeat the previous action, leading to low efficiency or even failure in completing the task (Sutton & Barto, 2018). This problem can be more destructive in high-frequency real-time control (HFRT) scenarios. For instance, in high-speed navigation scenarios, unmanned aerial vehicles must be frequently adjusted to ensure smooth flying and flexible obstacle avoidance. In computer games, the AI-controlled non-player characters (NPCs) may become stuck due to high latency in client interaction, thereby negatively impacting the gaming experience.

The conventional remedies for solving the delayed or fragmentary interaction issue include device restart, data recollection, bandwidth increasing and underlying network protocol optimization (Li et al., 2016). However, the shortcomings of such approaches are evident. Device restart causes the operating agent to lose control in real-time scenarios. Data recollection increases the time cost, resulting in low algorithmic efficiency. Besides, the recollected data can still be interrupted. Increas-

---

[1]The link failure can be caused by obstruction, insufficient bandwidth, switch failure, etc.

ing bandwidth will consume more manpower and material resources. The benefit of underlying network protocol optimization (e.g. transport layer protocol) is limited in a high delay network (Ahmed et al., 2003). Some recent works focus on recovering lost data by modeling the environmental dynamics (Li et al., 2014; Chen & Wu, 2015; Dong et al., 2009). However, in realistic control tasks, modeling the environmental dynamics is difficult due to the complexity of the environment. Besides, these methods require separate modeling of each scenario and thus lack generalization.

The most simple way to alleviate the delayed or fragmentary interaction issue is using 'action-repeat' (also commonly known as frameskip) (Kalyanakrishnan et al., 2021), where the same action (usually the last action) is repeated during a fixed interval. For example, in simple video games (e.g. Atari games (Braylan et al., 2015a)), appropriately setting action-repeat can simplify exploration and facilitate the learning of polices (Braylan et al., 2015b). However, improper action-repeat can impede exploration and lead to suboptimal policies. Besides, action repetition can leads to internal homogeneity of the action sequence and the inability to change the action at key states. Thus, the policy is unstable. Another way is to let the RL algorithms make up-front decisions (advance decision) for the future steps according to the current state or the received delayed state. Ramstedt & Pal (2019) and Ramstedt et al. (2020) consider real-time RL control problems with constant delays or random delays and predict the future action $a_{t+c}$ based on the received delayed state $s_t$.

Compared to frameskip, these methods can improve action variety. However, RL algorithms make advanced decisions by concatenating action sequences. This increases the difficulty of exploration and leads to unstable training (Chen et al., 2021). The stability of each method is analyzed in the appendix B.3

| Methods | Frameskip | advance decision | MARS |
|---|---|---|---|
| Generability | ✓ | ✓ | ✓ |
| stationarity | × | × | ✓ |
| Environment dynamic sensitivity | × | × | ✓ |

Table 1: A comparison on algorithmic properties of existing RL methods for fragmentary interactive control tasks

Furthermore, it is crucial for a general method to exhibit effectiveness across a wide range of environments. For example, in humanoid robot control, the changes in action are small to ensure balance. However, in the car navigation task, the range of changes in action can be very large in response to unexpected situations. Therefore, an ideal method should have dynamic sensitivity. As summarized in Table 1, none of the above methods is able to offer three desired properties, i.e., **generability, stationarity and environment dynamic sensitivity**, at the same time.

In this paper, we propose Multi-step action representation (MARS), which is the first DRL framework for solving fragmentary interactive control tasks while simultaneously achieving all three properties outlined in Tab.1. The high level idea is shown in Figure 1: MARS constructs a unified and decodable representation space for original multi-step

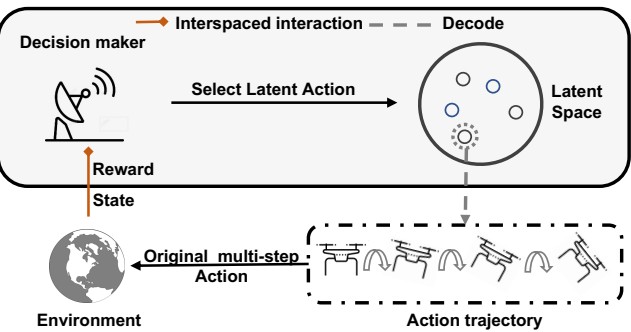

Figure 1: Conceptual overview of MARS.

actions, among which the agent learns a stable latent policy. Then, the selected latent action is decoded to the original action space so as to interact with the environment. MARS is inspired by recent advances in Representation Learning in DRL. Action representation learning has shown the potential to improve learning performance (Whitney et al., 2019) and hybrid action contorl (Li et al., 2021). MARS relies on a conditional Variational Auto-encoder (c-VAE) (Sohn et al., 2015) that conditions on the states and employs the embedding of dynamic transition between actions to construct the latent representation space for the associated multi-step actions (See Appendix A for complete preliminaries). The modular architecture of MARS is applicable to all reinforcement learning algorithms, which ensure the generalization of MARS. Besides, we use the unsupervised environmental dynamics to learn dynamics predictive multi-step action representation. Such a representation space can be semantically smooth, i.e., multi-step action representations that are close in the space have a similar influence on environmental dynamics. Moreover, to capture the dynamic transition semantics between multi-step actions, we propose the action transition scale,

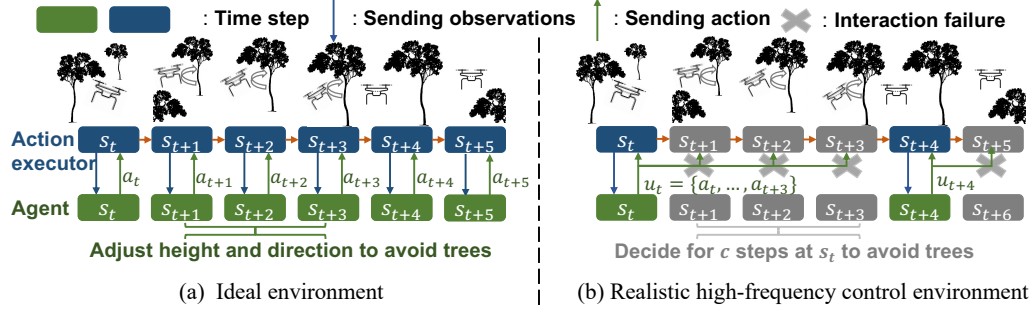

(a) Ideal environment

(b) Realistic high-frequency control environment

Figure 2: In this case, $c = 3$. The left picture is the decision-making process in the ideal scenario, and the right one is the fragmentary interaction phenomenon in the real-time scenario. How to make decisions for the unknown state is the key to solving this problem.

thus making MARS provide targeted latent action spaces for specific states. In our experiments, we evaluate MARS in a variety of environments with fragmentary interaction.

Our main contributions are summarized as follows: (1) We propose the first DRL framework for solving fragmentary interactive control tasks via multi-step action representation learning. (2) We provide an unsupervised method of learning a compact and decodable representation space for multi-step actions, along with two modules to improve the effectiveness of latent policy learning. (3) MARS consistently outperforms baselines in almost all related tasks, especially demonstrating significant superiority. When the environment becomes complex and the interactions are long-spaced. (4) MARS significantly improves the performance of real-world high-frequency robot control tasks based on fragmentary interaction.

## 2  PROBLEM FORMULATION

Interaction is an indispensable requirement for real-time control (especially high-frequency (HF) control tasks). As shown in Figure 2, the agent relies on interaction to obtain information of the environment and simultaneously sends actions to the executor.

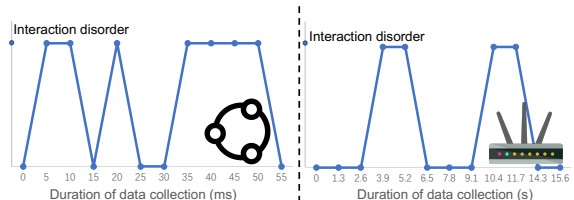

Figure 3: (a) Random fragmentary interaction caused by poor link contact. (b) Constant fragmentary interaction caused by the wireless card periodically scan.

Therefore, continuous interaction ensures continuity and efficiency of real-time control. However, we need to mitigate the impact of delay and packet loss on interactions in real-world. Even if the single-step transmission delay is alleviated through communication optimization methods (e.g., increasing bandwidth), the action execution delay is negligible in many scenarios due to the short execution time (especially in HF control tasks). Multi-step delay and packet loss still cause the current information untimely transmitted, reducing the interaction frequency. We model such real-time control problems as fragmented interactive reinforcement learning problems.

### 2.1  FRAGMENTARY INTERACTION ENVIRONMENTS

We formalize the fragmentary interaction reinforcement learning into four components: the agent, the unstable channel, the action executor, and the real-time environment. The agent needs to decide the appropriate action $a_t$ for each state $s_t$ in real-time to ensure efficient completion of the task. The challenge is that the interaction between the agent and the action executor is discontinuous. Because the packet delay and loss in the channel lead to sparsity in observations received by the agent. Besides, the states received by the action executor from the agent are also sparse due to fragmentary interactions. An illustration is shown in Figure 2, the fragmentary interaction problem is defined as follows: For a given observation $s_t$, how does the decision maker choose an action sequence $u$ that conforms to the decision logic of the corresponding unobserved state $\{s_{t+1}, ..., s_{t+c}\}$ (where $c$

denotes the interaction interval). Besides, Figure 3 shows that fragmentary interaction tasks can be divided into *constant* and *random* fragmentary interaction settings.

## 2.2 FRAGMENTARY INTERACTION MARKOV DECISION PROCESS (FIMDP)

We consider MDPs where the state space is continuous ($S \in \mathbb{R}^n$) and the action of each time step is $a \in A$. Different from ordinary MDP in which the agent decides an action $a$ according to the current state $s$, FIMDP requires the agent to evaluate the action sequence $u = a_t, ..., a_{t+c}$ according to the current state. Thus, FIMDP is defined as:

$$FIMDP(E, \pi) , \ E = (S, U, P, R) \tag{1}$$

$\pi$ is the policy, $E$ denotes the set of all information involved in the environment. $S$ denotes the collected states, where $\check{S}$ represent lost states. $U$ is the set of action sequence $u$ and $P$ is transition distribution. In FIMDP, reward is defined as $R : r(s_t) = \sum_{i=t}^{n=c} r(s_i)$, which means the rewards received by the agent are accumulated. Because the information in $c$ steps is lost. Policy $\pi$ takes current state to select action sequence $u = \{a_t, ..., a_{t+c}\}$. Thus, the environment transition function $\mathbf{K}$ is defined as:

$$\mathbf{K}(s_{t+c}|s_t, u_t) = p(s_{t+c}|s_t, u_t)\pi(u_t|s_t) \tag{2}$$

## 3 MULTI-STEP ACTION REPRESENTATION

As mentioned in previous sections, it is non-trivial for an RL agent to learn with fragmentary interaction efficiently due to the unobservability of intermediate states. Naive solutions, such as frameskip and advance decision, try to learn FIMDP policies directly by original reinforcement learning. However, these methods fail to provide the three desired properties: generability, stationarity and environment dynamic sensitivity (See Tab.1).

Inspired by recent advances in Representation Learning for RL (Chandak et al., 2019), we propose Multi-step action representation (MARS), a novel framework that converts the original multi-step action policy learning into a single-step policy learning problem. The intuition behind MARS is that the action of each step is heterogeneous in their original representations, but they jointly influence the environment. Thus, we assume that multi-step actions lie on a homogeneous manifold that is closely related to environmental dynamics semantics. In the following, we introduce an unsupervised approach to constructing a compact and decodable latent representation space to approximate such a manifold. The representation model is optimized by self-supervised technique. During the self-supervised training, transitions previously stored in the buffer or offline dataset are used. Besides, we find that MARS is not sensitive to data quality. In most scenarios, the data collected by random policies is used for effective training.

### 3.1 SCALE-CONDITIONED MULTI-STEP ACTION ENCODING AND DECODING

Although the vanilla VAE was used to construct the single-step action space in previous work, its representation of multi-step actions is of low quality. Because the concatenated dimension of the action sequence is high, it increases the difficulty of encoding and decoding. The encoder cannot represent the high-dimensional data in the latent space with high quality. The decoder cannot decode effectively according to the low-quality latent space variables. Thus, we propose the scale conditional VAE (sc-VAE) based on c-VAE for MARS. sc-VAE not only constructs the multi-step action latent space $z$, but also constructs the **action transition scale** space $\eta$. $\eta(u_t)$ describes the accumulation of action change scales for each action sequence $u_t$. sc-VAE regards $\eta$ as a priori condition. Encoder uses $\eta$ to guide the representation of multi-step actions. The decoder decodes precisely through the latent space variable and its corresponding $\eta$.

To realize differential evaluation of all action sequences at the same scale, we formula $\eta$ as follows:

$$\eta(u_t) = \sum \frac{\zeta_{u_t}}{(c-1) \times B} , \quad \zeta_{u_t} = \sum_{i=t}^{n=c-1} |a_{i+1} - a_i| \tag{3}$$

$c$ is the maximum interval and $B$ denotes the upper limit of action change. $\zeta_{u_t}$ denotes the sum of the absolute value of the difference between adjacent actions, which is used to evaluate the transition

scale of $u_t$. Note that $\zeta$ is also used as a regularization term to guide VAE optimizing (introduced later in this subsection). $\eta(u_t)$ normalizes $\zeta_{u_t}$ to $[0, 1]$.

For an action sequence $u_t$ and corresponding states $s_{t:t+c}$, our encoder $q_\phi(z|u_t, s_{t:t+c}, \eta(u_t))$ parameterized by $\phi$ takes $s_{t:t+c}$ and action transition scale $\eta(u_t)$ as conditions to build a multi-step action latent space, and maps action sequence $u_t$ into the latent variable $z \in \mathbb{R}^{d_1}$ ($d_1$ denotes the dimension of $z$). The decoder $p_\psi(\hat{u}_t|z, s_t, \eta(u_t))$ parameterized by $\psi$ then reconstructs the multi-step actions $u_t$ from $z$. During training, $\eta(u_t)$ can be obtained from buffer (or offline dataset) according to Eq.3. According to (Kingma & Welling, 2013), Variational Auto-Encoder (VAE) can be optimized by maximizing the variational lower bound. It should be emphasized that the input of the decoder is different from those of the encoder. Because in the FIMDP scenario, the intermediate states are missing. So we trained it to decode latent space action $z$ according to the current state $s_t$ and $\eta(u_t)$.

We employ a Gaussian latent distribution $N(\mu_x, \sigma_x)$ to model $q_\phi(z|u_t, s_{t:t+c}, \eta(u_t))$ where $\mu_x$ and $\sigma_x$ are the mean and standard deviation outputted by the encoder. The decoder decodes the latent variable $z \sim N(\mu_x, \sigma_x)$ as following: $\hat{u}_t = g_{\psi_1} \circ p_{\psi_0}(z, s_t, \eta(u_t))$, $g_{\psi_1}$ is a fully-connected layer for reconstruction. $B \circ A$ denotes function A's output acts as the input of function B. $p_{\psi_0}$ denotes the shared layers of the decoder. $\psi_{i \in \{1,2,3\}}$ denote the parameters of the prediction

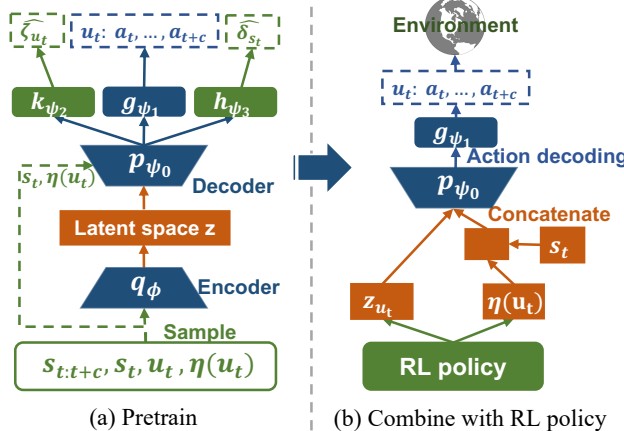

Figure 4: Detailed architecture of MARS.

networks. To ensure that the latent space learns the action transition scale, the decoder reconstructs $\hat{\zeta_{u_t}}$: $\hat{\zeta_{u_t}} = k_{\psi_2} \circ p_{\psi_0}(z, s_t, \eta(u_t))$. The loss function of sc-VAE is $L_{VAE}$:

$$L_{VAE}(\phi, \psi) = \mathbb{E}_{s,u \sim D, z \sim q_\psi} \big[ \|\hat{u}_t - u_t\|_2^2 + \|\hat{\zeta_{u_t}} - \zeta_{u_t}\|_2^2 \\ + D_{KL}\big(q_\phi(\cdot|u_t, s_{t:t+c}, \eta(u_t))\|N(0, I)\big) \big] \tag{4}$$

Where $D$ is the buffer. The first term is the reconstruction error (i.e., mean square error, MSE), the last term is the Kullback Leibler divergence $D_{KL}$ between the variational posterior of latent representation $z$ and the standard Gaussian prior. By using the reparameterization trick (Kingma & Welling, 2013), $\hat{u}_t$ is differentiable with respect to $\psi$ and $\phi$. For any latent variables $z_{u_t}$, they are decoded into multi-step actions $u_t$ conveniently by the VAE decoder. We summarize the encoding and decoding process:

$$\begin{aligned} \textbf{Encoder} :&z_{u_t} \sim q_\phi(\cdot|u_t, s_{t:t+c}, \eta(u_t)) \quad \forall\, s_{t:t+c}, u_t, \eta(u_t) \\ \textbf{Decoder} :&u_t = g_{\psi_1} \circ p_{\psi_0}(z_{u_t}, s_t, \eta(u_t)) \quad \forall\, s, z_{u_t}, \eta(u_t) \end{aligned} \tag{5}$$

## 3.2 Transition Aware Multi-step Action Representation

In the last section, we introduce how to build a scale-aware and decodable representation space for multi-step actions. However, it is still inefficient to learn the policy and value functions in the latent action space learned by the vanilla sc-VAE. Because the policy and value functions are highly dependent on the environmental dynamics. However, the vanilla VAE does not explicitly capture the effects of different multi-step actions on the environment (Grosnit et al., 2021). To address this issue, we further apply an unsupervised learning loss based on environmental dynamics prediction to further refine the multi-step action representation.

Ideally, the dynamics predictive representation should be semantically smooth, i.e., the action sequences decoded by the close points in the latent space should have a similar influence on the environment. Such a semantically smooth latent space is superior to the vanilla one in function approximation and generalization (Schwarzer et al., 2020).Based on this idea, MARS captures the

environmental dynamics by predicting the state transition residual. Besides, another advantage of using environment dynamics for represnetation clustering is that environment dynamics are more accessible and contain richer information than rewards or Q values in FIMDP. Detailled analysis see Appendix B.4. Note that MARS predicts the residual difference between the state after the execution of $u_t$ and the current state $s_t$. As shown in the left of Figure 4, $h_{\psi_3}$ is a subnetwork of our decoder. For any transition sample $(s_t, u_t, s_{t+c})$, the state residual is denoted by $\delta_{s_t} = s_{t+c} - s_t$. $p_{state} = h_{\psi_2} \circ p_\psi$. The predictions $\hat{\delta_{s_t}}$ is produced as:

$$\hat{\delta_{s_t}} = p_{state}(z_{u_t}, s_t, \eta(u_t)) \quad \forall s_t, z_{u_t}, \eta(u_t) \tag{6}$$

The environmental transition prediction loss is defined as:

$$L_{dy}(\phi, \psi) = \mathbb{E}_{s_t, u_t, s_{t+c}} \left[ \|\hat{\delta_{s_t}} - \delta_{s_t}\| \right] \tag{7}$$

Above all, the ultimate training loss for the multi-step action representation is denoted as follows:

$$L_{MARS}(\phi, \psi) = L_{VAE}(\phi, \psi) + \beta L_{dy}(\phi, \psi) \tag{8}$$

where $\beta$ is a hyper-parameter that controls the relative importance of the $L_{dy}$ and $L_{VAE}$. $L_{MARS}$ only depends on the environmental dynamic data which is reward-agnostic and is easier to obtain (Erraqabi et al., 2021; Yarats et al., 2021).

## 3.3 DRL WITH MULTI-STEP ACTION REPRESENTATION

As a plug-in method, MARS can be applied to any RL algorithm. MARS provides two action spaces for the RL algorithm: the multi-step action space $z$ and the action transition scale space $\eta$. RL algorithm maximizes the reward expectation by selecting optimal latent space actions from these two spaces. In this section, we apply MARS to a typical model-free RL method TD3 (Fujimoto et al., 2018) as an example. TD3 is a deterministic Actor-Critic algorithm. As illustrated in the right part of Figure 4, with the learned transition-aware multi-step action representation, the actor network learns a latent policy $\pi_\tau$ that outputs the latent actions, i.e., $[a_\eta, a_z] = \pi_\tau(s)$; $a_\eta \in \eta$, $a_z \in z$. Then we obtain the corresponding multi-step actions $u$ (in the original action space) by decoding the latent action $a_\eta$ and $a_z$ according to Eq.5.

The critic networks $Q_{\theta_1}, Q_{\theta_2}$ take the latent actions $a_z$ and $a_\eta$ as inputs to approximate the value function $Q_{\pi_\tau}$, i.e., $Q_{\theta_{i=1,2}}(s, a_\eta, a_z) \approx Q_{\pi_\tau}(s, a_\eta, a_z)$. We train the critic network using data sampled $(s, a_\eta, a_z, r, s')$. To make the expression of the formula clear, in this subsection $s$ is the current state. $s'$ is the first state of the next interaction cycle. The critic loss function is as follows:

$$L_Q(\theta_i) = \mathbb{E}_{s, a_\eta, a_z, s'} \left[ (y - Q_{\theta_i}(s, a_\eta, a_z))^2 \right] \text{ for } \forall i \in \{1, 2\} \tag{9}$$

Where $y = r + \gamma \min_{j=1,2} Q_{\bar{\theta}_j}(s', \pi_{\bar{\tau}}(s'))$, $\bar{\tau}$ denotes the target network parameters. The actor is updated according to the Deterministic Policy Gradient (Silver et al., 2014) as follows:

$$\nabla_\tau J(\tau) = \mathbb{E}_s \left[ \nabla_{\pi_\tau(s)} Q_{\theta_1}(s, \pi_\tau(s)) \nabla_\tau \pi_\tau(s) \right] \tag{10}$$

The overall algorithm MARS-TD3 is summarized in Algorithm 1, which contains two major stages: (1) the warmup stage and (2) the policy learning stage. In stage 1, MARS is trained using a prepared replay buffer $D$. The sc-VAE is updated by minimizing the VAE and the environmental dynamic prediction loss. Note that the proposed algorithm has no requirement on how the buffer $D$ is prepared and here we simply use a random policy to gather the data. In stage 2, given an environment state, the latent policy outputs the latent action $a_z$ and the action transition scale $a_\eta$ perturbed by the Gaussian exploration noise. The latent action is decoded into the original multi-step actions so as to interact with the environment, after which the collected transition sample is stored in the replay buffer $D$. Then, the latent policy is updated using the data sampled from $D$. The action representation model is updated concurrently in the second stage to make continual adjustments to the change of data distribution. The detailed network architecture is presented in appendix B.1. Appendix B.5 contains the way we guarante the training stability in FIMDP.

---

**Algorithm 1** MARS-TD3

---

Initialize actor $\pi_\tau$ and critic networks $Q_{\theta_1}, Q_{\theta_2}$
Initialize conditional VAE $q_\phi, p_\psi$ and replay buffer $D$

**Stage 1**

**while** not reaching warmup training times **do**
    Fill $D$ with data generated by random policy or offline datasets
    Update $q_\phi, p_\psi$ using samples in $D$
**end while**

**Stage 2**

**while** $t <$ policy training time **do**
    $a_\eta, a_z = \pi_\tau$ (with Gaussian noise)
    $u = g_{\psi_1} \circ p_{\psi_0}(a_z, s, a_\eta)$
    Execute $u$, observe $r$ and new state $s'$
    Fill $D$ with $(s, s_{t:t+c}, u, a_z, a_\eta, r, s')$
    Sample from $D$, update $Q_{\theta_1}, Q_{\theta_2}$ and $\pi_\tau$
    **if** reach representation training time **then**  Update $q_\phi, p_\psi$ using samples in $D$
    **end if**
**end while**

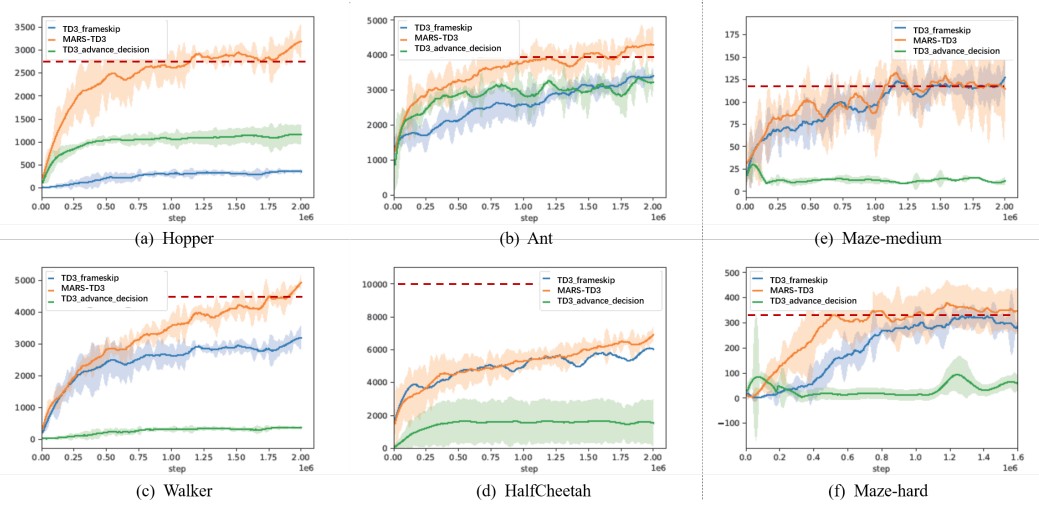

Figure 5: Comparisons of methods in constant fragmentary interaction tasks. Red lines denote the results of TD3 at the ideal setting with no interval. The x- and y-axis denote the environment steps and average episode reward over recent 10 episodes. The curve and shade denote the mean and a standard deviation over 5 runs.

# 4 EXPERIMENT

We empirically evaluate MARS to answer the following research questions. (1) **RQ1**: Can MARS effectively address the FIMDP problem and outperform related baselines in various *simulation* tasks? (2) **RQ2**: Can MARS help improve the performance of *real-world* robotic control problems based on fragmentary interaction? (3) **RQ3**: Can MARS be applied to all kinds of RL algorithms? (4) **RQ4**: Whether the action transition scale and the state dynamic prediction both play important roles in latent action optimization?

## 4.1 EXPERIMENTAL SETUPS

**Benchmarks.** To verify MARS in various aspects, we selected two different types of fragmentary interaction tasks: robot control (four openai Mujoco tasks (Brockman et al., 2016))and navigation.

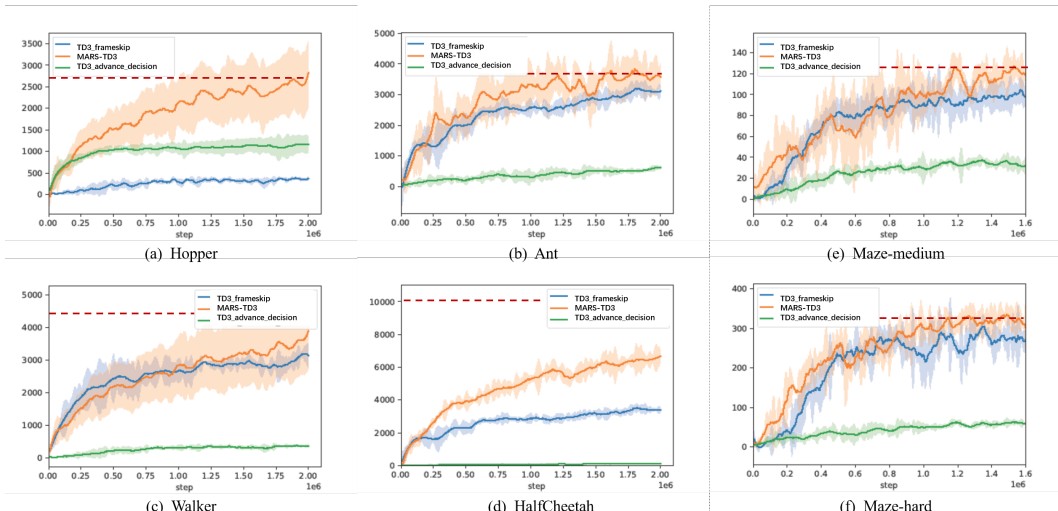

Figure 6: Results of random FIMDP tasks. The coordinates have the same meaning as Figure 5.

For navigation tasks, we used the medium and difficult maps of 2dmaze in D4RL (Fu et al., 2020), the agent's goal is to walk to the end of the maze (The original action dimension is low,i.e., 2).

**Baselines.** As far as we know, there is no specific solution to the FIMDP problem. Therefore, we designed two reasonable methods as baselines. The first is the RL algorithm combined with the frameskip trick, the second is using the RL algorithm to directly make decisions for the last $c$ steps (referred to simply as advance decision).

## 4.2 PERFORMANCE OF SIMULATION TASKS (RQ1)

To counteract implementation bias and achieve comprehensive comparison, all methods are implemented based on TD3 (Fujimoto et al., 2018) of the same architecture. For all tasks, we set latent action space dimension as 8 and environment dynamic predictive representation loss $\beta$ as 5. For the Mujoco tasks, the steps in the warm-up (stage 1) are 400000, and in the navigation task, the steps in stage 1 are 100000. Parameter setting can be found in appendix B.5. We first evaluate the performance in six environments of constant fragmentary interaction. In real-world tasks, an execution time for one step is about $0.05s$ to $0.25s$. And most of the prohibited interaction duration is $0.5s$ to $2s$. Thus, we set the interval step length as 8 to simulate the real-world setting. The results in Figure 5 show that MARS-TD3 outperforms other methods in all tasks, especially in the high-action dimension tasks (i.e., Mujoco). This proves that MARS can effectively solve the FIMDP problem, and avoid the convergence difficulties caused by dimensional explosion. Besides, although TD3-frameskip and TD3 advance decision can learn effective policies in navigation tasks, their learning efficiency is slow, shortages are more obvious in high-dimensional action tasks (mujoco). Further, MARS-TD3 is comparable to the ideal TD3 even in long-interval interaction settings, and even better on Hopper and Walker. This is because MARS compresses episode length with multi-step decision making, which reduces task difficulty and makes TD3 optimization easier.

We evaluate MARS on random FIMDP tasks, the above six environments are transformed into random fragmentary interaction setting. In the real-world, the longest interaction interval of the drone scene is generally $0.5s$ to $1.5s$, and the game delay is generally $0.1s$ to $1.5s$. Thus, we set the longest interval to 10 time steps. Figure 6 shows that MARS performs better than other baselines on all tasks. Compared with constant FIMDP, the effects of all methods are reduced to varying degrees in random fragmentary interaction tasks. However, MARS's scores are still the same as the ideal TD3 score in most tasks, but slightly fluctuating. The possible reason is that the agent may receive cumulative rewards after the execution of the two action sequences, which interferes with the evaluation of the latent space action. We will try to overcome this problem in the future.

## 4.3 PERFORMANCE OF REAL-WORLD ROBOTIC CONTROL TASK (RQ2)

We evaluate MARS in a real-world high-frequency snake robot control task. Snake robots are widely used in outdoor exploration because of their lightweight and flexible movements. However, the complex observation (i.e., 54-d) multi-joint (i.e., 24), redundant degrees of freedom make it hard to control. Visual description and detailed analysis is in Appnedix C.5. As a key of the control system, MARS solves the control task of the snake robot in the fragmentary interaction outdoor scene.

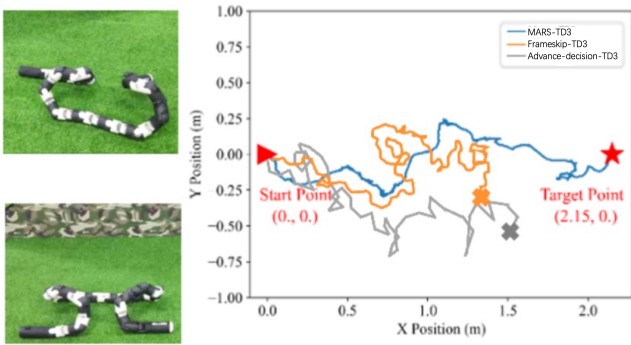

Figure 7: The left pictures are the snake robot posture. The right part is the path visualization.

The result is shown in figure 7. We will release the robot control system in the near future.

## 4.4 GENERALIZATION OF MARS (RQ3)

We test MARS with popular RL methods on 2dmaze and Mujoco. In summary, MARS can effectively combine with different RL methods. We analyze MARS's excellent generalization performance in detail based on experiments in appendix C.1.

## 4.5 ABLATION STUDY AND VISUAL ANALYSIS (RQ4)

We further evaluate the contribution of the major components in MARS: action transition scale $\eta$ and state dynamic prediction. Figure 8 (a) shows that both modules effectively optimize the latent space. when they are combined, the modeling ability of the environmental dynamic is improved. See the complete analysis and additional results in Appendix B.4. Figure 8 (b)

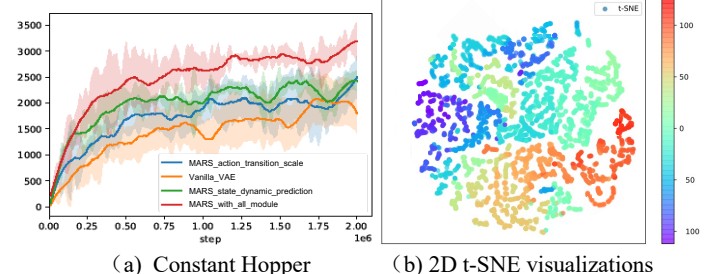

（a）Constant Hopper  （b）2D t-SNE visualizations

Figure 8: Ablation study, the curve and shade denote the mean and a standard deviation over 3 runs.

uses t-SNE (van der Maaten & Hinton, 2008) to visualize the learned latent action representations. We color each action based on its impact on the environment. result shows that actions with a similar impact on the environment are relatively closer in the latent space.), which demonstrates these two modules are helpful for deriving multi-step action representation.

Besides, results in Appendix C.3 improve the robustness of MARS to different interaction interval settings. Results in Appendix C.4 show the influence of latent space dimensions on MARS. Finally, the analysis of self-supervised training steps is in Appendix C.5.

## 5 CONCLUSION

In this paper, we propose Multi-step action representation (MARS) for RL methods to efficiently solve the fragmentary interaction tasks. MARS uses an unsupervised method to derive a compact and decodable representation space for multi-step actions. MARS can be easily combined with all kinds of DRL methods, making DRL algorithms learn effective policies. We evaluate MARS in a variety of environments. The results demonstrate the superiority of MARS when compared with baselines. Besides, MARS improves the performance of real-world HF robot control tasks based on fragmentary interaction.

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

## A  PRELIMINARIES

**Markov Decision Process (MDP).** A standard MDP can be represented as a tuple: $(\mathcal{S}, \mathcal{A}, \mathcal{P}, \mathcal{R}, \gamma, T)$, where $\mathcal{S}$ denotes the state set, $\mathcal{A}$ denotes an action set, $\mathcal{P}$ is the transition function: $\mathcal{S} \times \mathcal{A} \times \mathcal{S} \rightarrow [0, 1]$ and $\mathcal{R}$ is the reward function: $\mathcal{S} \times \mathcal{A} \rightarrow \mathbb{R}$. $\gamma \in [0, 1)$ is a discount factor and $T$ is the decision horizon. The target of the agent is to optimize its policy to maximize the expected discounted cumulative reward.

**Differences between FIMDP and POMDP, or MDP with reward delays.** The main difference between FIMDP and delay MDP: In FIMDP All environmental information is delayed (observation, action sequence, reward). Besides, agents are not allowed to pause midway. However, in reward delay MDP agent just need address the reward delay. And reward delay MDP does not consider the harm caused by the agent stalled in the middle. POMDP does not involve delay, the agent gets a local observation at each step. In contrast, the FIMDP has a delay in obtaining observations, but the agent can obtain global observations.

**Frameskip.** Frame-skipping may be viewed as an instance of (partial) open-loop control, under which a predetermined sequence of (possibly different) actions is executed without heed to intermediate states. Aiming to minimize sensing, Kalyanakrishnan et al. (2021) proposes a framework for incorporating variable-length open-loop action sequences in regular (closed-loop) control. The primary challenge in general open-loop control is that the number of action sequences of some given length $d$ is exponential in $d$. Consequently, the main focus in the area is on policies to prune corresponding data structures (Braylan et al., 2015b). Since action repetition restricts itself to a set of actions with size linear in $d$, it allows for $d$ itself to be set much higher in practice. With frame-skipping, the agent is only allowed to sense every $d$ state: that is, if the agent has sensed a state $s_t$ at time step $t >= 0$, it is oblivious to states $s_{t+1}, s_{t+2}, ..., s_{t+d-1}$, and next only observes $s_{t+d}$.

**Variational Auto-encoder.** The variational auto-encoder (VAE) is a directed graphical model with certain types of latent variables, such as Gaussian latent variables. A generative process of the VAE is as follows: a set of latent variable $z$ is generated from the prior distribution $p_\theta(z)$ and the data x is generated by the generative distribution $p_\theta(x|z)$ conditioned on $z : z \sim p_\theta(z), x \sim p_\theta(x|z)$. In general, parameter estimation of directed graphical models is often challenging due to intractable posterior inference. However, the parameters of the VAE can be estimated efficiently in the stochastic gradient variational Bayes (SGVB) framework, where the variational lower bound of the log-likelihood is used as a surrogate objective function. In this framework, a proposal distribution $q_\theta(x|z)$, which is also known as a "recognition" model, is introduced to approximate the true posterior $p_\theta(x|z)$. The multilayer perceptrons (MLPs) are used to model the recognition and the generation models. Assuming Gaussian latent variables, the first term of Equation (2) can be marginalized, while the second term is not. Instead, the second term can be approximated by drawing samples $z^{(l)}(l = 1, ..., L)$ by the recognition distribution $q_\theta(x|z)$, and the empirical objective of the VAE with Gaussian latent variables is written as follows:

$$L_{VAE}(\phi, \psi) = \frac{1}{L} \sum_\theta (x|z^{(l)}) - KL\big(q_\phi(z|x)||N(0, I)\big) \tag{11}$$

## B  EXPERIMENTAL DETAILS

### B.1  NETWORK STRUCTURE

| Layer | Actor Network | Critic Network |
|---|---|---|
| Fully Connected | (state dim, 256) | (statedim + $\eta$ dim + latent space dim, 128) |
| Activation | ReLU | ReLU |
| Fully Connected | (256, 128) | (256, 128) |
| Activation | ReLU | ReLU |
| Fully Connected | (128,latent space dim) and $\eta$ dim | (128, 1) |
| Activation | Tanh | None |

Table 2: Network Structures for DRL Methods

Our codes are implemented with Python 3.7.9 and Torch 1.7.1. All experiments were run on a single NVIDIA GeForce GTX 2080Ti GPU. Each single training trial ranges from 4 hours to 17 hours,

depending on the algorithms and environments. For more details of our code refer to the HyAR.zip in the supplementary results. And will open source code in the near future.

Our codes are implemented with Python 3.7.9 and Torch 1.7.1. All experiments were run on a single NVIDIA GeForce GTX 2080Ti GPU. Each single training trial ranges from 4 hours to 17 hours, depending on the algorithms and environments. For more details of our code refer to the HyAR.zip in the supplementary results. And will open source code in the near future.

Our TD3 is implemented with reference to `github.com/sfujim/TD3` (TD3 source-code). DDPG and PPO are implemented with reference to `https://github.com/sweetice/Deep-reinforcement-learning-with-pytorch`. For a fair comparison, all the baseline methods have the same network structure (except for the specific components of each algorithm) as our MARS-TD3 implementation. As shown in Tab.2, we use a two-layer feed-forward neural

| Model Component | layer | dimension |
|---|---|---|
| Conditional Encoder Network | Fully Connected (encoding) | $(\mathbb{R}^x, 256)$ |
| | Fully Connected (condition) | (stae dim + $\eta$ dim, 256) |
| | Element-wise Product | ReLU (encoding), ReLU(condition) |
| | Fully Connected | (256, 256) |
| | Activation | ReLU |
| | Fully Connected (mean) | (256, latent space dim) |
| | Activation | None |
| | Fully Connected (log std) | (256, latent space dim) |
| | Activation | None |
| Conditional Decoder, Prediction Network | Fully Connected (latent) | (latent space dim, 256) |
| | Fully Connected (condition) | (stae dim +$\eta$ dim, 256) |
| | Element-wise Product | ReLU (encoding), ReLU(condition) |
| | Fully Connected | (256, 256) |
| | Activation | ReLU |
| | Fully Connected ($\eta$) | (256, action dynamic transition) |
| | Activation | None |
| | Fully Connected (reconstruction) | (256, multi-step action dim) |
| | Activation | None |
| | Fully Connected | (256, 256) |
| | Activation | ReLU |
| | Fully Connected (prediction) | (256, state dim) |
| | Activation | None |

Table 3: Network structures for the Multi-step action representation (MARS).

network of 256 and 256 hidden units with ReLU activation (except for the output layer) for the actor network for all algorithms. For DDPG the critic denotes the Q-network. For PPO, the critic denotes the V-network. All algorithms (TD3, DDPG, PPO) output two heads at the last layer of the actor network, one for latent action and another for dynamic transition potential.

The structure of MARS is shown in Tab.3. We use element-wise product operation (Mahmood et al., 2018) and cascaded head structure (Fuchs et al., 2021) to our model.

## B.2 HYPERPARAMETER

For all experiments, we use the raw state and reward from the environment, and no normalization or scaling is used. No regularization is used for the actor and the critic in all algorithms. An exploration noise sampled from N(0, 0.1) (Dong et al., 2009) is added to all baseline methods when selecting an action. The discounted factor is 0.99 and we use Adam Optimizer (Li et al., 2016) for all algorithms. Tab.4 shows the common hyperparameters of algorithms used in all our experiments.

## B.3 THE ANALYSIS OF STABILITY

Section 1 of new version provides detailed analysis: Our method is more stable than frameskip and Advance decision.

The essence of frameskip is the repetition of an action, which leads to internal homogeneity of the action sequence and the inability to change the action at key states. Thus, the policy is unstable.

| Hyperparameter | TD3-frameskip | TD3-advance | MARS-PPO | MARS-TD3 | MARS-DDPG | |
|---|---|---|---|---|---|---|
| Actor Learning Rate | $1e^{-4}$ | $1e^{-4}$ | $1e^{-4}$ | $1e^{-4}$ | $1e^{-4}$ | $1e^{-4}$ |
| Critic Learning Rate | $1e^{-3}$ | $1e^{-3}$ | $1e^{-3}$ | $3e^{-4}$ | $3e^{-4}$ | $1e^{-3}$ |
| Representation Model Learning Rate | None | None | None | $1e^{-4}$ | $5e^{-3}$ | $5e^{-3}$ |
| Discount Factor | 0.99 | 0.99 | 0.99 | 0.99 | 0.99 | 0.99 |
| Batch Size | 128 | 128 | 128 | 128 | 128 | 128 |
| Buffer Size | $1e5$ | $1e5$ | $1e5$ | $1e5$ | $1e5$ | $1e5$ |

Table 4: A comparison of common hyperparameter choices of algorithms. We use 'None' to denote the 'not applicable' situation.

Advance decision needs to output the whole action sequence (concatenate c steps), and the long action sequence will increase the output dimension ($output\ dim = c \times single\ action\ dim$). This increases the difficulty of the action space exploration. Thus, the agent cannot learn the optimal policy.

Our method represents diverse action sequences into low-dimensional space. RL algorithms only need to learn policies in the latent action space. Our method reduces the difficulty of exploration and performs better.

### B.4 ANALYSIS OF REPRESENTATION CLUSTERING METHODS

The in-depth explanation is as follows (Section 1 of the new version, highlight) :

Get accurate Q value is difficult: In sparse reward environments (such as FIMDP), reward and Q are difficult to obtain and the evaluation of Q values in the early stage of training is inaccurate. In contrast, environmental dynamic is more reliable and accessible.

Environmental dynamic contains more information: The same reward or Q value may correspond to different environmental changes, but the same environmental change must have the same reward or Q value.

Environmental dynamic is reward-agnostic: In FIMDP, rewards are sparse. Environment dynamic do not require per-step reward. Therefore, environmental dynamic representation is more robust in FIMDP.

Further, in the following Table, we compare these three representational learning methods (1.cluster by Env dynamic 2. cluster by Q 3. cluster by reward). We only changed the clustering representations to ensure experimental fairness (buffer size is . Average of the 10 runs).

| Method | Halfcheetah | Walker | maze-hard |
|---|---|---|---|
| Env dynamic (ours) | $7012.1 \pm 131.4$ | $4821.6 \pm 427.6$ | $311.4 \pm 16.3$ |
| Q | $6386.1 \pm 412.7$ | $4021.6 \pm 313.7$ | $275.2 \pm 13.7$ |
| reward | $6618.1 \pm 372.7$ | $4188.3 \pm 185.5$ | $253.9 \pm 21.5$ |

Table 5: Average of the 10 runs

### B.5 HOW TO GUARANTE THE STABLE EXECUTING AND TRAINING

Section 3.2 of new version cover our method to guarante the training stability in detail (a common low-level method used in real-time control). We use the physical clocks on both devices to align. If the actions are obsolete, lose them. We retain the time stamp and execution flag of each action, which make actions executed in strict accordance with the timestamp order. When the new sequence arrives at the executor, the previous sequence will be replaced, and the execution flag of the unexecuted action will be False. The subsequent rewards will be accumulated into the new sequence. Thus, each latent space action reward is the sum of the executed action reward in the corresponding sequence. Following results shows the effect of alignment method (average of the 10 runs, Interval is 6).

| Method | Walker | maze-hard |
|---|---|---|
| Ours | $4463.2 \pm 362.7$ | $311.4 \pm 16.3$— |
| without aligment | $4168.3 \pm 372.6$ | $213.1 \pm 16.7$ |

Table 6: Average of the 10 runs, Interval is 6.

### B.6 ADDITIONAL IMPLEMENTATION DETAILS

For PPO, the actor network and the critic network are updated every 2 and 10 episode respectively for all environments. The clip range of the PPO algorithm is set to 0.2 and we use GAE (Sutton & Barto, 2018) for a stable policy gradient. For DDPG, the actor network and the critic network is updated at every 1 environment step. For TD3, the critic network is updated every 1 environment step and the actor network is updated every 2 environment steps.

The default latent action dim is 8, We set the KL weight in representation loss $L_{MARS}$ as 0.5. Environment dynamic prediction loss weight $\beta$ is 5 (default).

## C ADDITIONAL EXPERIMENT

### C.1 GENERALIZATION OF MARS

We test MARS with popular RL methods on three tasks: Hopper, Walker, and hardMaze. To make the experiment fair, we used the same parameters for all methods and implemented them based on public code. We use each RL algorithm to train on three tasks under the ideal setting and compare them with their corresponding improvement methods. To show the optimal score after the algorithm convergence, we train all the algorithm's 2000000 time steps. The results in Tab.7 show that all methods can learn effective policies with the help of MARS and perform similarly to their ideal settings. The differences in scores are mainly due to the variation in performance of the RL algorithms. In summary, MARS can be combined with different methods to provide a reliable action space for solving FIMDP as normal MDP with RL.

| Benchmarks | MARS-PPO | MARS-DDPG | MARS-TD3 |
|---|---|---|---|
| Maze hard | 256 \| 0.7 ↑ | 243 \| 2.5 ↑ | 311 \| 16.3 ↑ |
| Hopper | 2811.4 \| 73.5 ↓ | 1815.6 \| 184.3 ↑ | 3384 \| 53.1 ↑ |
| Walker | 3831.2 \| 285.1 ↓ | 1032.7 \| 201.9 ↓ | 4821.6 \| 427.6 ↑ |

Table 7: MARS generalization verification. All tasks are set to constant FIMDP, interval is 8. The format of the data in the table is: MARS-RL score | the score gap. ↓ denotes score of MARS lower than the ideal setting baseline. ↑ denotes score of MARS higher than the ideal setting baseline. All scores are averaged over five runs.

### C.2 DETAILS OF ABLATION STUDY

We conducted two experiments to show how well the two mechanisms of MARS work together. Although the results of randomized FIMDP and constant FIMDP are slightly different, the same conclusion can be derived: The green curves in Figure 9 demonstrate that the representation model with increased action transition scale is much better than the original VAE. This means that dynamic transition potential can create an action hidden space by explicitly modeling the dependence between multi-step actions. The blue curves also show that VAE with state dynamic prediction is better than the original VAE because it can represent action sequences that have similar environmental effects at close locations. Finally, the red curves show that the two mechanisms work well together in MARS, and combining them improves representation ability.

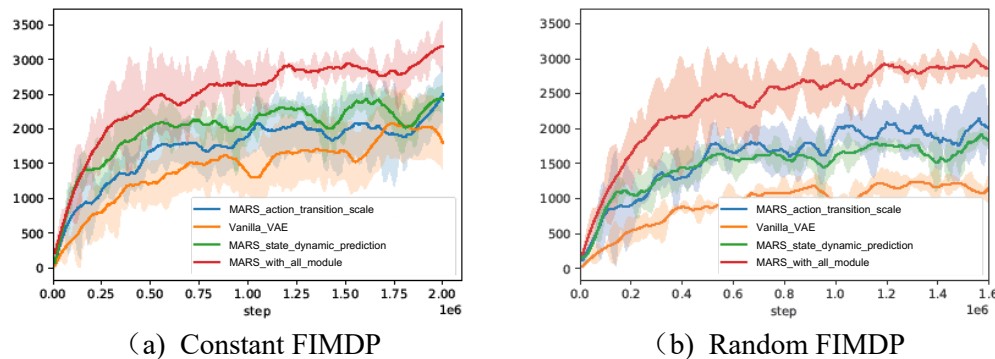

(a) Constant FIMDP   (b) Random FIMDP

Figure 9: Details of ablation study. The curve and shade denote the mean and a standard deviation over 5 runs.

## C.3 VALIDITY VERIFICATION OF MULTIPLE INTERACTION INTERVALS

To further demonstrate the effectiveness of MARS in diverse fragmentary interaction scenarios. For constant fragmentary interaction control tasks, we uniformly set the forbidden interaction duration and conducted four experiments on Hopper. The results in Figure 10 show that MARS can solve most tasks effectively and still guarantee good scores at long intervals, but the effectiveness of MARS decreases significantly when the interval is too long (which is not common in real-world scenarios). We believe that this is because VAE is unable to effectively characterize excessively long sequences, leading to the failure of multi-step action space modeling.

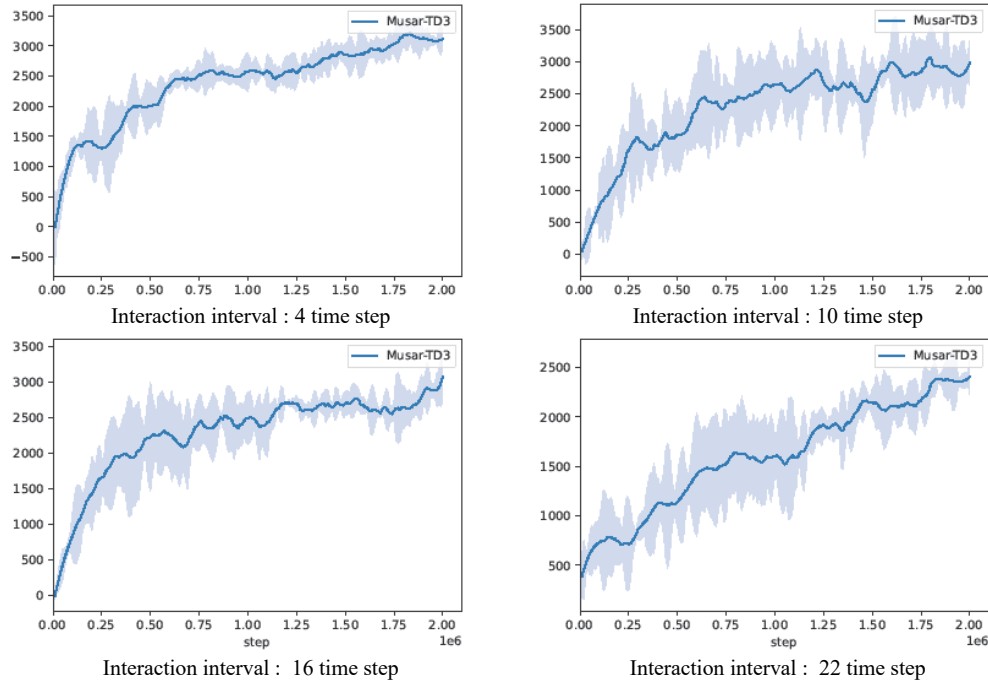

Figure 10: MARS's experimental results under four different settings of forbidden interaction duration. The curve and shade denote the mean and a standard deviation over 5 runs.

In addition, to observe the sensitivity of MARS to interaction intervals on random FIMDP tasks, we uniformly set the forbidden interaction duration and conducted four experiments on Hopper. The results in Figure 11 show that in random FIMDP scenarios, MARS performs well in both short and medium-interval scenarios. However, convergence changes slowly in the very long interval scenario, and the score is only half that of the medium interval task. Because MARS's representational capabilities are not perfect for modeling long action sequences for extremely long-spaced tasks (even if this setting rarely occurs in real-world scenarios). Therefore, in the future, we hope to find more suitable representation models to overcome this problem.

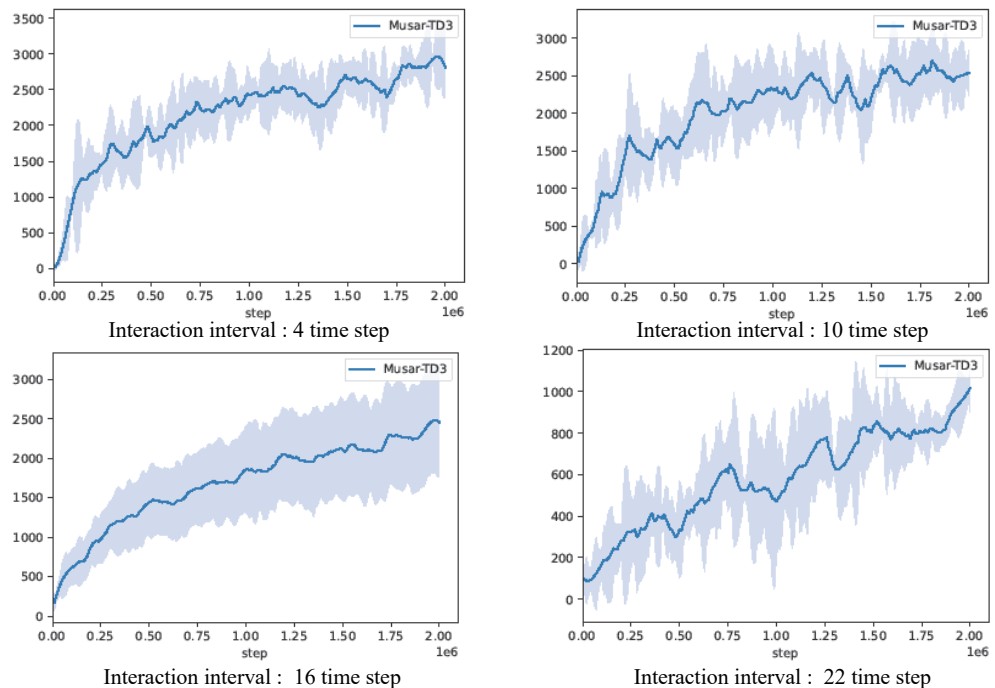

Figure 11: MARS's experimental results under four different settings of forbidden interaction duration. The curve and shade denote the mean and a standard deviation over 5 runs.

## C.4 THE INFLUENCE OF LATENT ACTION SPACE DIMENSION ON ALGORITHM EFFECT

The representation space dimension of VAE is an important hyperparameter. If the latent space dimension is too low, a large amount of original data information will be lost, resulting in invalid representation space. On the contrary, when the latent space dimension is too large, the calculation amount of the model will be increased. To verify the sensitivity of MARS to latent space dimensions, we test it on two tasks with different original action dimensions.

We set up four sets of latent space dimensions for constant FIMDP Hopper (interaction interval time step: 8, original action dimension: 3, so the action sequence dimension to be modeled is 24). The learning curve in Figure 12 shows that for raw data of such high dimensions, when the latent space dimension is set too low, the latent space information will be lost, resulting in the convergence failure of reinforcement learning policies. On the contrary, too high a latent space dimension increases the complexity of reinforcement learning policy exploration. In addition, we set up four comparison experiments on the 2dmaze task with a lower dimension of the original action sequence (interaction interval time step: 4, original action dimension: 2, so the action sequence dimension to be modeled is 8). The experimental results in Figure 13 show that the suboptimal policy can be learned when the latent space dimension is low, because the original data dimension is low. So the low-dimensional latent space loses less information. The score increases as the latent space dimension increases.

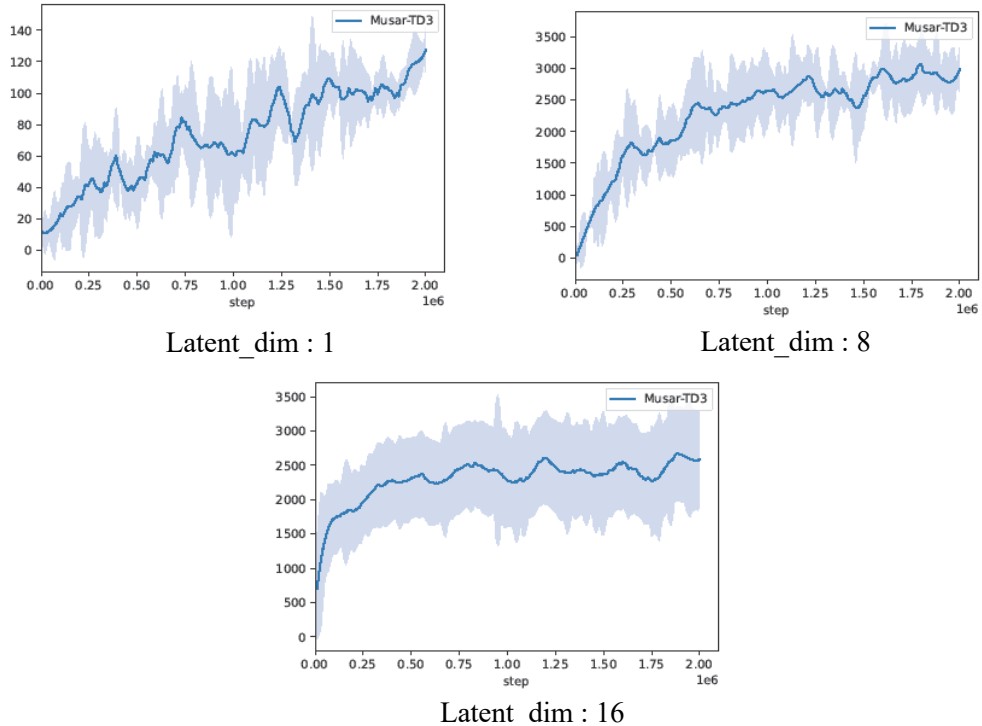

Figure 12: TD3 learning curves of three different latent space dimensions set the corresponding. The curve and shade denote the mean and a standard deviation over 5 runs.

However, when the latent space dimension is too high, the score will drop significantly, which is because of the exploration difficulties brought by high-dimensional latent space.

## C.5    THE INFLUENCE OF ENVIRONMENT STEPS OF WARMUP STAGE

In this section, we conduct some additional experimental results for a further study of MARS from different perspectives: We provide the exact number of samples used in the warm-up stage (i.e., stage 1 in Algorithm 1 in each environment in Tab.8. The number of warm-up environment steps is about $5\% \sim 10\%$ of the total environment steps in our original experiments. Moreover, we also conducted some experiments to further reduce the number of samples used in the warm-up stage (at most $80\%$ off). See the colored results in Tab.8. MARS can achieve comparable performance with $< 3\%$ samples of the total environment steps.

Conclusion: The number of warm-up environment steps is about $5\% \sim 10\%$ of the total environment steps in our original experiments. The number of warmup environment steps can be further reduced by at most $80\%$ off (thus leading to $< 3\%$ of the total environment steps) while the comparable performance of our algorithm remains.

Snake robot point tracking is a periodic information acquisition task. The reason for this phenomenon is not packet loss or delay, but the unique rolling gait of the snake robot. The rolling gait of the snake robot is the most efficient gait for outdoor work at present, which includes the rolling of the head sensor. However, this gait causes the next observation to be acquired only when the head sensor rolls to a position parallel with the ground. If the rolling motion is interrupted midway, the next state cannot be obtained, leading to agent ineffective shaking. Our control system is the first to complete the rolling gait tracking task. We describe the snake robot tracking task in detail and provide several visualization results in the new version. The snake robot needs to scroll from the starting point to the target point in a scene with a size of 5 square meters. Each observation step is 54d, the action sequence time step is 20, so the action sequence dimension is 1.08k. And the control

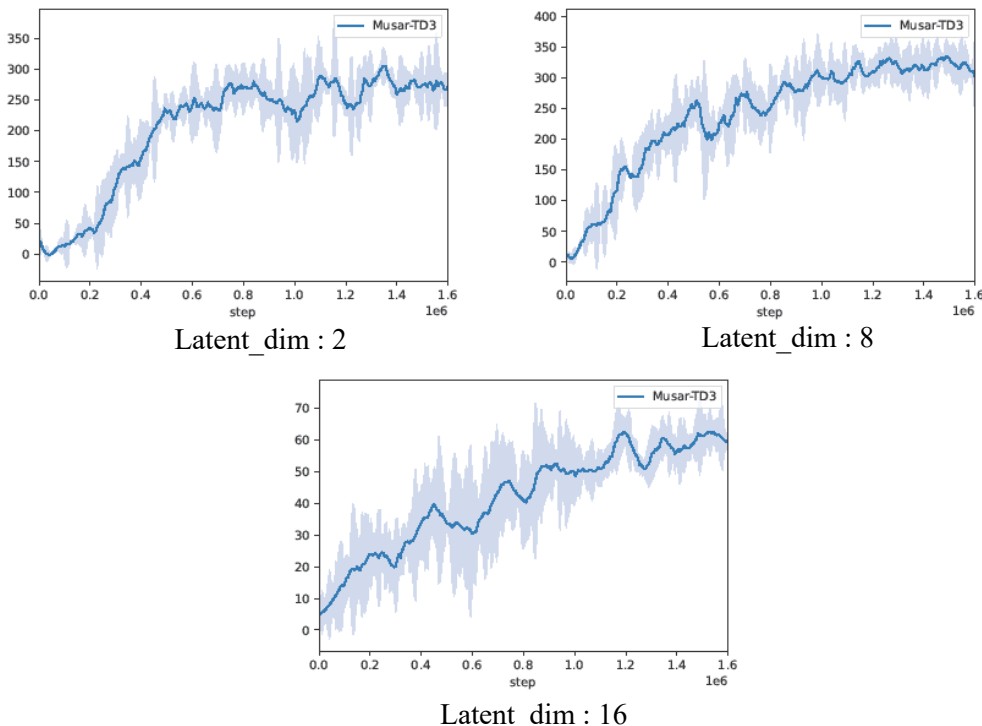

Figure 13: TD3 learning curves of three different latent space dimensions set the corresponding. The curve and shade denote the mean and a standard deviation over 5 runs.

| Environment | Warm-up steps (original) | Warm-up steps (new) | Total Env. Steps |
|---|---|---|---|
| Hopper | 400000(0.08\|3219.1) | 100000(0.02\|3086.4) | 5000000 |
| Ant | 400000(0.08\|4305.7) | 100000(0.02\|4025.6) | 5000000 |
| Walker | 400000(0.08\|4961.3) | 100000(0.02\|4792.6) | 5000000 |
| HalfCheetah | 400000(0.08\|6593.2) | 100000(0.02\|6071.2) | 5000000 |
| 2dmaze-medium | 100000(0.083\|127.8) | 30000(0.025\|118.5) | 1200000 |
| 2dmaze-hard | 100000(0.083\|327.6) | 35000(0.0292\|296.1) | 1200000 |

Table 8: The exact number of samples used in warm-up stage training in different environments. The column of 'original' denotes what is done in our experiments; the column of 'new' denotes additional experiments we conduct with fewer warm-up samples (and proportionally fewer warm-up training). For each entry $x(y|z)$, x is the number of samples (environment steps), y denotes the percentage number of $\frac{warm-up\ environment\ steps}{number\ of\ total\ environment\ steps\ during\ the\ training\ process}$, and z denotes the corresponding performance of MARS-TD3.

frequency of the real interaction is 20 Hz. Here is the demo video and several visual description of the task: https://anonymous.4open.science/r/ICLR2024-C0F6/README.md.

## D  MARS-DDPG PSEUDOCODE

---

**Algorithm 2** MARS-DDPG

---

Initialize actor $\pi_\tau$ and critic networks $Q_\theta$
Initialize conditional VAE $q_\phi, p_\psi$ and replay buffer $D$
**Stage 1**
  **while** *not reaching warmup training times* **do**
    Fill $D$ with data generated by random policy or offline datasets
    Update $q_\phi, p_\psi$ using samples in $D$
  **end while**
**Stage 2**
  **while** $t < policy\ training\ time$ **do**
    $a_z, a_\eta = \pi_\tau$ (with Gaussian noise)
    $u = p_\psi(a_z, a_\eta, s)$
    Execute $u$, observe $r$ and new state $s'$
    Fill $D$ with $(s, s_{t:t+c}, u, a_z, a_\eta, r, s')$
    Sample from D, update $Q_\theta$ and $pi_\tau$
    **if** reach representation training time **then**Update $q_\phi, p_\psi$using samples in $D$

---

