# OpenReview forum: "Addressing Real-Time  Fragmentary Interaction Control Problems via Muti-step Representation Reinforcement Learning"
_ICLR.cc/2024/Conference — Submitted to ICLR 2024_

### Official Review · Reviewer_XM6T · 2023-10-29

**Soundness:** 2 fair
**Presentation:** 2 fair
**Contribution:** 2 fair
**Rating:** 5
**Confidence:** 2

**Summary:**

The paper proposes a method for real-time control scenarios in which interation between the execution devices and the computation node is lossy. The question here is what actions should be taken by the execution devices when the latest inference result has not arrive yet. The algorithm involves planning a sequence of future actions instead of a single action and learning a latent representation for lists of future actions in an unsupervised manner. Then an RL policy is trained to solve the task with the action space being the latent representation. Experimental results illustrate the effectiveness of using latent representations as action spaces in both simulation environments and the real world.

**Strengths:**

1. The paper is well-written and easy to follow.
2. The motivation is clear and the method is clean.

**Weaknesses:**

1. The experiemnt part may not fully match with the motivation of the paper. Generally speaking, simulation environments such as Mujoco do not require the use of framentary control.
2. The real-world robotic control experiment lack important details, including the interaction pattern between the executor and the agent in the real-world experiment.

**Questions:**

1. Why does the method significantly outperform TD3 with advanced decision? Please explain the comparison between the baselines in more details.
2. What is the interaction pattern between the executor and the agent in the real-world experiment?

---

> ### Author Response · Authors · 2023-11-19
> **Thank you for your valuable comments. Hope that our answers can address your concerns.**
>
> # Response
> Thank you for your valuable comments. Hope that our answers can address your concerns.
> ## The experiment part may not fully match the motivation of the paper. Generally speaking, simulation environments such as Mujoco do not require the use of fragmentary control.
> Fragmented interactions exist in many virtual scenarios. For example, remote control of NPCS: In the game, the cloud server needs to control the terminal NPC in real time. NPC stalling can significantly degrade the user experience. So we follow works with related evaluation scenarios (e.g.RTAC[1]RDAC[2]). We used mainstream virtual scenarios (Mujoco, D4RL) to simulate remote control of NPC tasks instead of using Mujoco directly. The details of constructing fragmented interactions simulation tasks are described in the experiment section Sec 4.2: In real-world tasks, an execution time for one step is about 0.05s to 0.25s. And most of the prohibited interaction duration is 0.5s to 2s. Thus, we set the interval step length as 8 to simulate the real-world setting.  We will highlight the task settings in the new version. And put it at the top of the experiment section.
>
> [1] Chris Pal, et al. "Real-time reinforcement learning." NeurIPS (2019).
>
> [2] Ramstedt, Simon, et al. "Reinforcement learning with random delays." arXiv preprint arXiv:2010.02966 (2020).
> ## The real-world robotic control experiment lacks important details, including the interaction pattern between the executor and the agent in the real-world experiment.
> The 21-joint snake robotic uses a rolling gait. The algorithm is arranged in the cloud server, and the decision end interacts with the snake robot through the local area network. [This is the interaction pattern figure.](https://anonymous.4open.science/r/ICLR2024-C0F6/interaction_pattern.png)
>
> Snake robot tracking is a periodic information acquisition task. The reason for this phenomenon is not packet loss or delay, but the unique rolling gait of the snake robot. This gait of the snake robot is the most efficient gait for outdoor work, which includes the rolling of the head sensor. However, this gait causes the next observation to be acquired only when the head sensor rolls to a position parallel to the ground. If the rolling motion is interrupted midway, the next state cannot be obtained, leading to ineffective shaking of the robot.
>
> The snake robot needs to scroll from the starting point to the target point in a scene with a size of 5 square meters. Each observation step is 54d, and the action sequence time step is 20, so the action sequence dimension is 1.08k. The control frequency of the real interaction is 20 Hz. We describe the snake robot tracking task in detail and provide several [visualization results](https://anonymous.4open.science/r/ICLR2024-C0F6/) in the new version.
> ## Why does the method significantly outperform TD3 with advanced decision?
> Explanation:
> - Td3-advance decision needs to output the whole action sequence (concatenate c steps), and the long action sequence will increase the output dimension (output_dim=c * single_action_dim). This increases the difficulty of the action space exploration. Thus, the agent cannot learn the optimal policy (mentioned in the introduction).
> - Our method represents diverse action sequences in low-dimensional space. Our method reduces the difficulty of exploration and performs better.
> - Ours is a plug-in method and can be applied to all major reinforcement learning methods. To ensure experimental fairness, all methods use Td3. This eliminates the framework advantages that Td3 brings to advance deisicion.
>
> Verified by experiment:
> - Following results show the sensitivity of three methods to the action sequence dimension. The first table runs on Walker and the second runs on Maze-hard.
> - The exploration ability of Td3-advance decision decreases significantly with the increase of sequence length.
> - TD3-frameskip has better exploration ability than TD3- advanced decision. This is because frameskip reduces the spatial dimension of exploration by sacrificing action variety (repeating the same action).
> - Our method has the best performance by reducing the dimension of exploration space and ensuring the diversity of action through the representation of action space.
>
> | method     | 12 dim| 24dim| 36 dim|
> | :-----------: | :-----------: | :------------: | :-----------: |
> | Ours |271.2 ± 11.4|258.3 ±15.2|246.5 ± 21.3|
> | TD3- frameskip |231.2 ± 14.6|112.1 ± 3.2|88.7 ± 2.5|
> | TD3- advanced decision  |83.5 ± 5.7|31.9 ± 7.4|85.1 ± 2.8|
>
> | method     | 18 dim| 36 dim| 54 dim|
> | :-----------: | :-----------: | :------------: | :-----------: |
> | Ours |4715.6 ± 343.1|4613.2 ± 362.7|4215.9 ± 428.3|
> | TD3- frameskip |3714.7 ± 252.1|941.6 ± 603.2|691.3 ± 122.4|
> | TD3- advanced decision |2368.2 ± 316.4|913.2 ± 592.1|718.3 ± 176.8|
>
> ## If our reply addresses your concerns, we would appreciate it if you could kindly consider raising the score.

---

> ### Author Response · Authors · 2023-11-22
> **Please allow us to analyze your question again**
>
> ## We think that the supplementary experiment is inadequately described. Please Allow us to analyze it again.
> ## Why does the method significantly outperform TD3 with advanced decision?
> Explanation:
> - Td3-advance decision needs to output the whole action sequence (concatenate c steps), and the long action sequence will increase the output dimension (output_dim=c * single_action_dim). This increases the difficulty of the action space exploration. Thus, the agent cannot learn the optimal policy (mentioned in the introduction).
> - Our method represents diverse action sequences in low-dimensional space. Our method reduces the difficulty of exploration and performs better.
> - Ours is a plug-in method and can be applied to all major reinforcement learning methods. To ensure experimental fairness, all methods use Td3. This eliminates the framework advantages that Td3 brings to advance deisicion.
>
> Verified by experiment:
> - Following results show the sensitivity of three methods to the action sequence dimension. The first table runs on Walker and the second runs on Maze-hard. Walker's single-step action dimension is 3. To analyze the sensitivity of the methods to the action space dimension, we set the number of decision steps as follows: 4 (dim = $4\times3$), 8 (dim = $8\times3$), 12 (dim = $12\times3$). Maze-hard's single-step action dimension is 2. This navigational task requires the agent to change its action in a specific state (e.g., change the directionto to avoid running into a wall). Excutor with actions repeating hard to get to the end of the maze, and therefore fail to get the maximum reward resulting in an inaccurate Q-value fit. Thus, when the action sequence of the policy is internally homogeneous,  long decision steps lead to training instability. We set the number of decision steps as follows: 9 (dim = $9\times2$), 18 (dim = $18\times2$), 27 (dim = $27\times3$).
> - The exploration ability of Td3-advance decision decreases significantly with the increase of sequence length.
> - TD3-frameskip has better exploration ability than TD3- advanced decision. This is because frameskip reduces the spatial dimension of exploration by sacrificing action variety (repeating the same action).
> - Our method has the best performance by reducing the dimension of exploration space and ensuring the diversity of action through the representation of action space.
>
> | method     | 12 dim| 24dim| 36 dim|
> | :-----------: | :-----------: | :------------: | :-----------: |
> | Ours |271.2 ± 11.4|258.3 ±15.2|246.5 ± 21.3|
> | TD3- frameskip |231.2 ± 14.6|112.1 ± 3.2|88.7 ± 2.5|
> | TD3- advanced decision  |83.5 ± 5.7|31.9 ± 7.4|85.1 ± 2.8|
>
> | method     | 18 dim| 36 dim| 54 dim|
> | :-----------: | :-----------: | :------------: | :-----------: |
> | Ours |4715.6 ± 343.1|4613.2 ± 362.7|4215.9 ± 428.3|
> | TD3- frameskip |3714.7 ± 252.1|941.6 ± 603.2|691.3 ± 122.4|
> | TD3- advanced decision |2368.2 ± 316.4|913.2 ± 592.1|718.3 ± 176.8|

---

> > ### Comment · Area_Chair_T9tP · 2023-11-22
> > **Please respond to the author reply**
> >
> > Dear reviewer, please do respond to the author reply and let them know if this has answered your questions/concerns.

---

> > ### Author Response · Authors · 2023-11-23
> > **We further provide detailed real-world experiment visual descriptions of device information and interaction patterns**
> >
> > We further provide detailed [real-world experiment visual descriptions of device information and interaction patterns.](https://anonymous.4open.science/r/ICLR-AB36/Detailed%20interaction%20pattern%20with%20all%20the%20device%20information.png)

---

### Official Review · Reviewer_cdnx · 2023-10-30

**Soundness:** 3 good
**Presentation:** 3 good
**Contribution:** 2 fair
**Rating:** 5
**Confidence:** 4

**Summary:**

The paper proposes a method for solving real-time control tasks where the delay or the loss of the network packets may lead to fragmentary interaction. Specifically, this could happen when the remote controller fails to receive the observation on time and thus can not issue a new command to the robot. Without receiving correct commands at the correct timestep, the robot may standstill by doing nothing or repeating the last action. Both may induce a failure in task completion.

To address this, the paper proposes to generate an action sequence instead of a single action step when making decisions. Thus when the remote controller can not produce the new action sequence due to failing to receive the new observation, the robot can still realize what to do according to the remaining commands in the action sequence received last time. This is a bit like how traditional planning algorithms such as MPC work.

To achieve this, a modified VAE is trained to construct a latent action space, which serves as the action space for RL methods like TD3. Every time the actor-network chooses an action from the latent action space, the latent variable will be converted to a robot command sequence through the decoder.

In the experiments, several Mujoco environments are constructed to simulate the fragment interaction situation. The results show that the proposed method can overcome this problem and compete with the agents trained and deployed in an ideal environment. Besides, a robot snake experiments are conducted to show it can be applied to real robots.

**Strengths:**

1. Investigating how to build a system robust to fragmentary interaction or latency is important in the robotics system.
2. The paper is easy to understand.
3. The Mujoco experiments show that the method works well and is comparable to baselines in an ideal environment. The ablation study shows the importance of different modules

**Weaknesses:**

The main weakness of this paper is the poor robot evaluations. As the main motivation of the method is to address a practical issue in the real-world robot learning environment, a comprehensive real-world evaluation should be conducted on a platform where the fragmentary interaction problem indeed exists and is critical to the robot's performance.

The paper only contains a short section about the snake robot experiment with simple proprioceptive observations. In this setting, the delay or the loss of the network packets rarely happens as the bandwidth should be enough for transmitting the small amount of data consisting of only 54-dimensional vectors without any high-dimensional images and lidar results. Also, if the snake fails to receive any commands, simply stopping by doing nothing and waiting for the new commands is acceptable. It is not like a legged robot, which may easily fall down if it can not receive a stable command stream.

In a word, my main concern is that it is not verified on robot platforms that indeed suffer from this problem like quadrupedal robots with multi-modal perception. Otherwise, I cannot believe the proposed method can solve the claimed problem.

**Questions:**

As a robotics-related submission, it is usually good to include demo videos. Sometimes, it is even more important than the paper itself. Could you please share more visualization results of the snake experiment?

---

> ### Author Response · Authors · 2023-11-19
> **Thanks for your valuable comments. Hope that our answers will address your concerns.**
>
> # Response
> Thanks for your valuable advice on robot learning. Hope that our answers will address your concerns.
> ## Details of snake robot experiment.
> Snake robot point tracking is a periodic information acquisition task. The reason for this phenomenon is not packet loss or delay, but the unique rolling gait of the snake robot. The rolling gait of the snake robot is the most efficient gait for outdoor work at present, which includes the rolling of the head sensor. However, this gait causes the next observation to be acquired only when the head sensor rolls to a position parallel to the ground. If the rolling motion is interrupted midway, the next state cannot be obtained, leading to agent-ineffective shaking.  Our control system is the first to complete the rolling gait tracking task. We describe the snake robot tracking task in detail and provide several [visualization results](https://anonymous.4open.science/r/ICLR2024-C0F6/) in the new version. The snake robot needs to scroll from the starting point to the target point in a scene with a size of 5 square meters. Each observation step is 54d, and the action sequence time step is 20, so the action sequence dimension is 1.08k. The control frequency of the real interaction is 20 Hz.
>
> In addition to robot scenarios, fragmented interactions also exist in many virtual scenarios. For example, remote control of NPCS: In the game, the cloud server needs to control the terminal NPC in real-time. NPC stalling can significantly degrade the user experience. So we follow works with related evaluation scenarios (e.g.RTAC[1]RDAC[2]). We used mainstream virtual scenarios (Mujoco, D4RL) to simulate remote control of NPC tasks. Our method outperforms in these tasks.
>
> [1] Chris Pal, et al. "Real-time reinforcement learning." Advances in neural information processing systems 32 (2019).
>
> [2] Ramstedt, Simon, et al. "Reinforcement learning with random delays." arXiv preprint arXiv:2010.02966 (2020).
> ## It is usually good to include demo videos.
> Thanks for your advice. We are willing to provide the visualization results, we provide a [demo video and several visual descriptions](https://anonymous.4open.science/r/ICLR2024-C0F6/) of the task in the new version.
> ## If our reply addresses your concerns, we would appreciate it if you could kindly consider raising the score.

---

> > ### Comment · Reviewer_cdnx · 2023-11-21
> > **Thank you**
> >
> > Thank you for the response.
> >
> > I checked the videos but could not get a clear conclusion. It is just a moving robot snake without anything informative. A demo video at least is supposed to:
> >
> > 1) Put a start point mark and a destination mark on the ground to indicate what the task is
> > 2) Have a baseline without the proposed method. The result should be that with the proposed method, the snake manages to reach the destination.
> >
> > The robotics research is demo-centric. Due to the limited real-world robot evaluation, I would like to keep my score unchanged.

---

> > > ### Author Response · Authors · 2023-11-22
> > > **Thanks for your thorough and courteous reply. We fully understand your concern. Please forgive us for bothering you again.**
> > >
> > > Thank you for your reply. We fully understand your concern. Please forgive us for bothering you again.
> > >
> > > We would like to provide detailed video results. However, most of our videos are required to be pinted with the institution logo and could not be released during the double-blind review period. In order to show the robot task as much as possible, we capture one of the frames and mask the logo to describe the task (in the anonymous link). All videos will be released after the review.
> > >
> > > The deep learning algorithm proposed in this paper is an important part of the robot snake control system. The complete control system and detailed robotic experiments will be expanded into a paper and submitted to a non-blind reviewed journal of field of robot learning. The robot snake experiment in this submission is more like a preview of the complete control system.
> > >
> > > This paper is biased towards focusing on providing a general algorithm to address FIMDP for the deep reinforcement learning community. To increase the visibility and recognition of our algorithm in the DRL community, we choose the mainstream simulation environments of the DRL community as the main experiment environments. And our method outperforms baselines on these environments. Besides, based on these tasks, we provide a large number of analytical ablation experiments. These experiments have the ability to prove the effectiveness of our method.
> > >
> > > Thank you for your thorough and courteous feedback. We hope that our response can change your perspective and alleviate your concerns.

---

### Official Review · Reviewer_GKKU · 2023-11-01

**Soundness:** 3 good
**Presentation:** 4 excellent
**Contribution:** 3 good
**Rating:** 6
**Confidence:** 3

**Summary:**

The paper proposed a representation learning method for reinforcement learning to handle real-time fragmentary interaction control problems. The authors proposed a novel problem formulation in the MDP, where the interaction between agents and environments might be fragmentary.The agents need to make multi-step decisions on potentially insufficient observation to handle the frame skip and package loss. The authors proposed a VAE-based approach to learn the multi-step latent representation and use the representation with RL to handle the fragmentary interaction problem. Empirical results have shown the effectiveness compared to intuition-based baselines.

**Strengths:**

1. The problem formulation is novel and significant. Fragmentary interaction is indeed an important problem in real-world high-frequency control problems.
2. The presentation is excellent, the problem formulation is clear and multiple figures help clarify the problems.
3. The proposed algorithm is solid and performs well empirically.

**Weaknesses:**

1. The authors might need to connect more with existing problem formulations. The FIMDP looks related to partially observable MDPs and MDP with reward delays. I can get a rough sense that there are differences between FIMDP and these existing problem formulations, but not very clear. The authors should add a clear discussion to distinguish FIMDP from the existing related problem formulations.

**Questions:**

1. The questions are also related to weakness, what are the differences between FIMDP and POMDP, or MDP with reward delays?
2. If we have/learned a world model for the environment, can we do the model-based predictions, like model predictive control to solve the FIMDP? (this might be the most straightforward method that first came into my mind.) How does this compare to learning the multi-step representations?

It is a good paper. I will consider improving my score if the questions are appropriately addressed.

---

> ### Author Response · Authors · 2023-11-19
> **Thanks for your recognition of our work. We hope our reply can address all your concerns.**
>
> # Response
> Thanks for your recognition of our work. We hope our reply can address all your concerns.
> ## The questions are also related to weakness, what are the differences between FIMDP and POMDP, or MDP with reward delays?
> We compare FIMDP with reward delayed MDP and POMDP in the new version:
> - The main difference between FIMDP and delay MDP: In FIMDP All environmental information is delayed (observation, action sequence, reward). Besides, agents are not allowed to pause midway. However, in reward delay MDP agents just need to address the reward delay. And reward delay MDP does not consider the harm caused by the agent stalled in the middle.
> - POMDP does not involve delay, the agent gets a local observation at each step. In contrast, the FIMDP has a delay in obtaining observations, but the agent can obtain global observations
>
> ## 2. If we have/learned a world model for the environment, can we make the model-based predictions, like model predictive control to solve the FIMDP? (this might be the most straightforward method that first came into my mind.) How does this compare to learning the multi-step representations?
> This question is valuable. In fact, we originally wanted to build a world model to achieve comprehensive environmental perception. However, the interval of FIMDP is too long and some tasks have random intervals. It is difficult to construct effective dynamics models and reward models under delayed conditions, which can be summarized as we cannot model random delays effectively. Therefore, we chose a method that is easier to train and robust to delay: construct an action latent space. To prove that our approach is more efficient than the model-based approach in FIMDP tasks. The following results show the performance of the original MBPO[1] based world model method (Average of the 10 runs. Interval is 10). In the future, we will further think about how to make Model-based methods effective in FIMDP tasks.
>
> | Methods     | Walker| maze-hard| HalfCheetah|
> | :-----------: | :-----------: | :------------: | :-----------: |
> | MBPO-based |3781.2 ± 217.5|203.6 ± 15.8|6153.1 ± 381.2|
> | Ours  |4715.6 ± 343.1|271.2 ± 11.4|7012.1 ± 131.4|
>
> [1] Janner, Michael, et al. "When to trust your model: Model-based policy optimization." Advances in neural information processing systems 32 (2019).
> ## If you think the above response addresses your concerns, we would appreciate it if you could kindly consider raising the score.

---

> > ### Comment · Area_Chair_T9tP · 2023-11-22
> > **Please respond to the author reply**
> >
> > Dear reviewer, please do respond to the author reply and let them know if this has answered your questions/concerns.

---

> > ### Comment · Reviewer_GKKU · 2023-11-23
> >
> > Thank you for the detailed response!
> >
> > I have a complex feeling about the paper.
> >
> > On the one hand, I like the paper because this is a realistic problem in the real world. It hasn't been studied according to the literature review. The methods proposed look good. Accepting this paper definitely helps motivate more people to focus on this problem.
> >
> > On the other hand, after reading other reviewers' comments, the experimental results are indeed not strong enough to sell the methods. The video is also not very helpful.
> >
> > I finally decided to keep my score, It is already a paper above the accept threshold. If the authors could further polish the paper and provide stronger empirical results, this work would have a broader impact.

---

### Official Review · Reviewer_3YNf · 2023-11-01

**Soundness:** 2 fair
**Presentation:** 3 good
**Contribution:** 2 fair
**Rating:** 6
**Confidence:** 3

**Summary:**

This work primarily focuses on real-time reinforcement learning for high-frequency robot control tasks, where the information transmission is not entirely reliable. The communication between the action executor and agent in reinforcement learning may be affected by packet loss and latency, potentially impacting the effectiveness of policy execution. In contrast to previous methods that directly generate multi-step action sequences, this paper proposes using sc-VAE to generate an intermediate representation in place of an action sequence. During actual execution, this intermediate representation is used to generate the corresponding action sequence. The paper introduces additional regularization for the influence of actions on the environment within the generated intermediate representation.

The proposed method's performance is tested in various Mujoco task environments and a real-world snake robot control task. The results indicate that MARS outperforms the advanced decision method that directly generates action sequences and a simple frame-skip method which makes decisions with lower frequency. Further ablation studies confirm that the proposed method can consider the influence of the environment when generating intermediate representations.

**Strengths:**

The strengths of this work are as follows:

1. The paper provides a detailed introduction to the background of the real-time RL problem, and the research objectives are clear.

2. The proposed method in the paper exhibits excellent generability and can work with various reinforcement learning optimization algorithms.

3. The paper offers experimental results on real robots, demonstrating the practicality of the proposed method.

**Weaknesses:**

The weaknesses of this work are as follows:

1. The soundness of the paper is limited. The method is based on sc-VAE, and the primary claim that "the action sequences decoded by the close points in the latent space should have a similar     influence on the environment" relies on empirical evidence and lacks theoretical explanation (refer to question 1).

2. The paper lacks explanations for some critical aspects of the experiments. For more details, please refer to question 2.

**Questions:**

1. Why is clustering representations of actions that have similar environmental effects better than clustering action sequences with similar values or rewards? Can you provide a more in-depth explanation and analysis?

2. In the random FIMDP tasks mentioned in the paper, is the number of decision steps fixed within one trial or randomly decided during execution? As shown in Figure 11, a larger interval leads to lower performance, how will the method perform if trained with longer action sequences but executed with a shorter interval, compared with training with shorter action sequences?

3. The paper mentions that MARS has better stationarity, but it doesn't provide relevant explanations and proofs. Additionally, in a real-time RL setting, where it's not guaranteed that the actions actually executed by the executor strictly match the policy's output, do the collected trajectory samples inherently lack stationarity? (you can regard the trajectory as being sampled from a rapidly changing environment transition probability)

4. I noticed some interesting results. Why does it appear that MARS has a more pronounced advantage over frame-skips in simpler tasks than in more complex tasks?

5. The application form of the method needs further clarification. Does the decoder need to be run on the execution device? If so, does this mean that latent representations will also be lost?

6. Should the "Musar" method in Fig.5 and 6 be referred to as "MARS"?

---

> ### Author Response · Authors · 2023-11-19
> **Thanks for the objective and in-depth comments. Hope that our reply can ease your concerns.**
>
> # Response
> ## Why is clustering representations of actions that have similar environmental effects better than clustering action sequences with similar values or rewards?
> Explanation is as follows (highlight in new version):
> - Get accurate Q value is difficult: In sparse reward environments (such as FIMDP), reward and Q are difficult to obtain and the evaluation of Q values in the early stage of training is inaccurate. In contrast, environmental dynamic is more reliable and accessible.
> - Environmental dynamic contains more information: The same reward or Q value may correspond to different environmental changes, but the same environmental change must have the same reward or Q value.
> - Environmental dynamic is reward-agnostic: In FIMDP, rewards are sparse. Environment dynamics do not require a per-step reward. Therefore, environmental dynamic representation is more robust in FIMDP.
>
> Further, we compare these 3 representational learning methods. We only changed the clustering representations.
>
> |Method|Halfcheetah|Walker|maze-hard|
> | :- | :- | :- | :- |
> |Env dynamic (ours)|7012.1±131.4|4821.6±427.6|311.4±16.3|
> |Q|6386.1±412.7|4021.6±313.7|275.2±13.7|
> |reward|6618.1±372.7|4188.3±185.5|253.9±21.5|
> ## Is the number of decision steps fixed within one trial? How will MARS perform if trained with longer action sequences but executed with a shorter interval, compared with training with shorter action sequences?
> The number of decision steps is fixed. The following results show that reducing the number of steps as much as possible can improve scores (Average of the 10 runs. Interval is 6). As the number of steps increases, both the sequence dimension and the reward sparsity increase. These lead to the difficulty in exploration.
> |task|6 step|12 step|18 step|
> | :-| :-| :- | :- |
> |Walker|4715.6±343.1|4613.2±362.7|4215.9±428.3|
> |maze-hard |271.2±11.4|258.3±15.2|246.5±21.3|
> ## How to guarantee that the actions actually executed by the executor strictly match the policy's output, do the collected trajectory samples inherently lack stationarity?
> We use the physical clocks on both devices(a common method in real-time control, hightlght in the new version). If the actions are obsolete, lose them. We retain the time stamp and execution flag of each action, which makes actions executed in strict accordance with the timestamp order. When the new sequence arrives at the executor, the previous sequence will be replaced, and the execution flag of the unexecuted action will be False. Each latent space action reward is the sum of the executed action reward in the corresponding sequence. Results show the effect of the alignment method (average of the 10 runs, Interval is 6).
>
> |Method|Walker|maze-hard|
> | :- | :- | :- |
> |Ours|4463.2±362.7|311.4±16.3|
> |without alignment|4168.3±372.6|213.1±16.7|
> ## MARS has a more advantage over frame-skips in simple tasks than in complex tasks?
> Because the VAE hyperparameters of MARS are not optimized in difficult tasks. We adjust the VAE hyperparameters while keeping the remaining hyperparameters unchanged for both methods （following table). By doing this MARS has a more pronounced advantage on complex tasks. (Old score of MARS in maze-Hard:315.2±13.9 maze-Hard:6417.3±317.3)
>
> | method| maze-hard|HalfCheetah|
> | :- | :- | :-|
> |Ours|356.4±16.3|8631.5±265.2|
> |TD3- frameskip|285.7±12.5|5842.3±306.7|
> ## MARS has better stationarity.
> Explanation:
> - Frameskip leads to internal homogeneity of the action sequence and the inability to change the action at key states. Thus, the policy is unstable.
> - Advance decision needs to output the whole action sequence (concatenate c steps), which will increase the output dimension. This increases the difficulty of the action space exploration. Thus, the agent cannot learn the optimal policy.
> - Our method represents diverse action sequences in low-dimensional space. RL algorithms only need to learn policies in the latent action space. Our method reduces the difficulty of exploration and performs better.
>
> Further, we compare the policy stability of all methods for sequence length. Ours provides high stability.
> | method|6 step|12 step|18 step|
> | :- | :-| :- | :- |
> |Ours|4715.6±343.1|4613.2±362.7|4215.9±428.3|
> |TD3- frameskip|3714.7±252.1|941.6±603.2|195.7±72.5|
> |TD3- advanced decision|2368.2±316.4|913.2±592.1|718.3±176.8|
> ## Does the decoder need to be run on the execution device? Does this mean that latent representations will also be lost?
> We do not deploy the decoder to the executor. The agent makes c step decision based on each timestep information, so each time step (t+i) will receive c times in the future. Besides, our decision steps are set at maximum intervals to ensure that the executor receives the next sequence before the previous sequence is completed.
> ## Should the "Musar" be referred to as "MARS"?
> Correct this in the new version.
>
> ## If our reply addresses your concerns, we would appreciate it if you could kindly consider raising the score.

---

> > ### Comment · Area_Chair_T9tP · 2023-11-22
> > **Please respond to the author reply**
> >
> > Dear reviewer, please do respond to the author reply and let them know if this has answered your questions/concerns.

---

> > ### Comment · Reviewer_3YNf · 2023-11-22
> > **Reply for the author**
> >
> > I appreciate the detailed response from the authors. The author's replies addressed some of my concerns and clarified certain ambiguities in the paper. After reviewing the questions from other reviewers and the author's responses, I have slightly and cautiously raised my score. My understanding of this paper is that the author effectively improves exploration and learning efficiency by learning a more compact latent representation rather than the original action sequence, as opposed to directly optimizing the action sequence. The focus of the paper is, in fact, slightly different from the core issue of real-time RL. Looking forward, in addition to addressing the mentioned points in the rebuttal, the author could consider whether placing the decoder on the executor side is feasible. Additionally, encoding key future actions in the action sequence could be explored to ensure the preservation of critical actions even in the event of communication loss. This would align more with real-time specified methods.

---

> > > ### Author Response · Authors · 2023-11-23
> > > **Thanks for your recognition of this work and your valuable suggestions**
> > >
> > > ## Response
> > >
> > > ##  The author could consider whether placing the decoder on the executor side is feasible.
> > > This question is very valuable. Our method allows the decoder to be placed in the executor. Because based on the technique [1] introduced earlier in response to your q3 and q5, we can ensure that at least one interaction is implemented within the maximum latency interval. This allow the excutor can maintain movement even when there is a delay. So The decoder can be put into the executor as long as the executor's processor has enough memory.
> > >
> > >  [1] We set the length of the decision sequence as the maximum delay length to ensure that the actor can keep moving during the period of no interaction, and enable the agent to make sequential decisions according to each state and use the time stamp to align with the decision end to achieve seamless connection between actions.
> > >
> > > ## Additionally, encoding key future actions in the action sequence could be explored to ensure the preservation of critical actions even in the event of communication loss.
> > > This suggestion is reasonable. We will explore this direction in the future
> > >
> > > ## Thanks for your recognition of this work and your valuable suggestions. We will continue to improve our ICLR submission based on your questions and suggestions.

---

### Meta-Review · Area_Chair_T9tP · 2023-12-05

**Metareview:**

(a) the method deals with situations where packets and communication may be dropped so an agent cannot be purely markovian and instead needs to anticipate and execute some actions without communication. They do this by learning a multi-step VAE that is called the sc-VAE. The sc-VAE essentially models the distribution of open loop actions given state snippets and action accumulations. They integrate this with deep RL methods and they then show its efficacy on tasks in simulation, a snake robot and a robot arm.

(b) the setting is very practical for robotics and the method,while a bit heuristic does seem effective in simulation. The real robot experiments are a plus!

(c) the experiments that *actually* need this seem to be contrived, the sim experiments clearly don't need this and the real world experiments are not very well described. Given so much is specific to the problem setting and it is non standard, that makes it hard to calibrate the significance of the results.

(d) better and more thorough descriptions/comparisons in the real world. more clear motivation and more realistic comparison perhaps in the real-world RL benchmark [Dulac-Arnold et al] or other domains.

**Justification For Why Not Higher Score:**

The experiments that *actually* need this seem to be contrived, the sim experiments clearly don't need this and the real world experiments are not very well described. Given so much is specific to the problem setting and it is non standard, that makes it hard to calibrate the significance of the results. The reviewers agreed, describing the fact that the method is somewhat heuristic and not very well motivated.

**Justification For Why Not Lower Score:**

N/A

---

### Decision · Program_Chairs · 2024-01-16

Reject